# Magnetic Molecularly Imprinted Polymers for the Rapid and Selective Extraction and Detection of Methotrexatein Serum by HPLC-UV Analysis

**DOI:** 10.3390/molecules27186084

**Published:** 2022-09-18

**Authors:** Tingting Zhou, Ziwen Deng, Qing Wang, Hui Li, Shun Li, Xuanming Xu, Yusun Zhou, Shukai Sun, Chao Xuan, Qingwu Tian, Limin Lun

**Affiliations:** 1Department of Clinical Laboratory, The Affiliated Hospital of Qingdao University, #No. 1677, Wutaishan Road, Qingdao 266000, China; 2Department of Blood Transfusion, Qingdao Women and Children’s Hospital, Qingdao 266000, China

**Keywords:** methotrexate, magnetic molecularly imprinted polymers, magnetic molecularly imprinted solid-phase extraction, high-performance liquid chromatography–ultraviolet detection

## Abstract

In this work, novel selective recognition materials, namely magnetic molecularly imprinted polymers (MMIPs), were prepared. The recognition materials were used as pretreatment materials for magnetic molecularly imprinted solid-phase extraction (MSPE) to achieve the efficient adsorption, selective recognition, and rapid magnetic separation of methotrexate (MTX) in the patients’ plasma. This method was combined with high-performance liquid chromatography–ultraviolet detection (HPLC–UV) to achieve accurate and rapid detection of the plasma MTX concentration, providing a new method for the clinical detection and monitoring of the MTX concentration. The MMIPs for the selective adsorption of MTX were prepared by the sol–gel method. The materials were characterized by transmission electron microscopy, Fourier transform-infrared spectrometry, X-ray diffractometry, and X-ray photoelectron spectrometry. The MTX adsorption properties of the MMIPs were evaluated using static, dynamic, and selective adsorption experiments. On this basis, the extraction conditions were optimized systematically. The adsorption capacity of MMIPs for MTX was 39.56 mgg^−1^, the imprinting factor was 9.40, and the adsorption equilibrium time was 60 min. The optimal extraction conditions were as follows: the amount of MMIP was 100 mg, the loading time was 120 min, the leachate was 8:2 (*v*/*v*) water–methanol, the eluent was 4:1 (*v*/*v*) methanol–acetic acid, and the elution time was 60 min. MTX was linear in the range of 0.00005–0.25 mg mL^−1^, and the detection limit was 12.51 ng mL^−1^. The accuracy of the MSPE–HPLC–UV method for MTX detection was excellent, and the result was consistent with that of a drug concentration analyzer.

## 1. Introduction

Methotrexate (MTX) is a dihydrofolate reductase inhibitor that acts on the G1 and G1/S transition cells to prevent DNA and RNA synthesis [1]. As a cell-cycle-specific antimetabolic drug commonly used in combined chemotherapy regimens, MTX is widely used in the treatment of pediatric acute lymphoblastic leukemia, lymphoma, chorionic epithelioma, osteosarcoma, and other malignant tumors [2]. Unfortunately, MTX can kill normal cells during chemotherapy because it lacks selectivity to tumor cells [3]. Therefore, side effects due to the increasing application of MTX, such as bone marrow suppression, neurotoxicity, and renal tubule obstruction and injury, have attracted the attention of clinicians [4]. These side effects are cause by the high MTX blood concentration in patients and the long administration of a high MTX dose [5]. Fortunately, DNA and RNA syntheses can be carried out normally through the exogenous administration of calcium leucofolate after MTX application [6]. Therefore, monitoring the MTX blood concentration and the timely application of calcium leucofolate can effectively reduce these side effects.

Current MTX detection methods include UV spectroscopy [7], capillary electrophoresis [8,9], mass spectrometry [10], homogeneous enzyme immunoassay, and fluorescence polarization immunoassay [11]. Although these methods can be used to determine the blood concentration of MTX, some drawbacks limit their use in clinical application [12]. In addition to the commonly used detection methods, the use of molecularly imprinted polymers (MIPs) for drug detection has also increased [13,14]. As a new molecular recognition technology, molecular imprinting combines materials science, polymer science, chemical engineering, biochemistry, and other disciplines [15]. Compared with the natural molecular recognition system, synthetic MIPs have the advantages of good stability, strong resistance to harsh environments, a low preparation cost, a long service life, renewability, and a wide application range [16]. Therefore, it has good application prospects in the fields of natural product separation, food detection, bionic sensing, solid phase extraction, antibodies, and receptor simulation [15,16]. MIPs are 3D network polymers formed by template molecules, monomers, and cross-linking agents initiated by initiators in pore-forming agents [17]. After elution of the template, MIPs leave imprinted holes that match the size and functional groups of the template molecules [18]. Therefore, MIPs have selectivity and predetermined adsorption on the target object and show a specific recognition ability that is comparable to biological antibodies [18]. In addition, MIPs also have excellent characteristics, such as a simple synthesis, low cost, high stability, and reproducibility, and thus show good application prospects in the field of small-molecule target recognition [19]. Martins et al. used condensed tannin as the template and a catechin standard solution to prepare MIPs and used them as adsorption materials for the solid-phase extraction of condensed tannins [13]. Zhou et al. used teicoplanin as the template to prepare MIPs, which were used as selective materials to construct a sensitive method based on molecularly imprinted solid-phase extraction combined with liquid chromatography–tandem mass spectrometry to determine the presence of vancomycin and norvancomycinin milk samples [14]. However, the separation process of MIPs from a solution is time consuming and involves the loss of target molecules, which may hinder the application of MIPs [20].

Magnetic molecularly imprinted solid-phase extraction (MSPE) combines magnetic substances, such as Fe_3_O_4_, with other molecularly imprinted solid-phase extraction technologies [21,22]. It uses specific imprinted pores as binding sites to improve the selectivity of the extraction process and to overcome interference in the complex matrix [23,24]. Importantly, it can quickly separate the polymer by magnetic action, reduce the loss of liquids and solids in the separation process, and further improve the enrichment coefficient of the target owing to the large surface area of the polymer nanoparticles (NPs) [25,26,27]. High-performance liquid chromatography (HPLC)–ultraviolet (UV) detection is considered to be a technology with a high stability and specificity for the detection of organic molecules in complex biological substrates [28,29]. Therefore, the combination of MSPE and HPLC–UV can achieve rapid selective separation and accurate determination of different components in complex matrices [30].

In this study, MSPE and HPLC–UV were combined to establish a reliable and accurate detection method for plasma MTX levels to simplify the sample pretreatment process and reduce the interference of proteins and other components for MTX detection.

## 2. Materials and Methods

### 2.1. Reagents

MTX (99%) and sulfamethoxazole (98%) were supplied by Macklin Biochemical Co., Ltd. (Shanghai, China). Trimethoprim (99%) was supplied by Brinway Biotechnology Co., Ltd. (Shanghai, China). Metsulfuronmethyl (98%) was purchased from Aladdin (Shanghai, China). 3-Aminopropyl triethoxysilane (APTES, 98%) and tetraethyl orthosilicate (TEOS, 98%) were obtained from Sigma-Aldrich (St. Louis, MO, USA). Analytical pure of methanol, ethanol, acetonitrile, ammonium hydroxide, potassium hydroxide, FeCl_3_·6H_2_O, and FeSO_4_·7H_2_O were obtained from Sinopharm Chemical Reagent Co., Ltd. (Shanghai, China). Acetic acid was purchased from Yantai Far East Fine Chemical Co., Ltd.

### 2.2. Instruments

The precision electronic balance was purchased from Sartorius (Goettingen, Germany). The air dry oven was obtained from Yiheng Scientific Instrument Co., Ltd. (Shanghai, China). The thermostatic water bath box was supplied by Tuan Mechanical and Electrical Automobile Equipment Co., Ltd. (Changsha, China). The CNC circular shaker was supplied by Scilocgex (Pittsburgh, PA, USA). The motor stirrer was purchased from Meiyingpu Instrument Manufacture Co., Ltd. (Shanghai, China). The HPLC–UV instrument and C18 chromatographic column were obtained from Waters (Milford, MA, USA). The transmission electron microscope (JEM 1200EX) was obtained from Electron Optics Laboratory (Tokyo, Japan). The Fourier transform infrared spectrometer was purchased from Bruker (karlsruhe, Germany). The X-ray diffractometer was supplied by Panalytical (Almelo, The Netherlands). The X-ray photoelectron spectrometer was obtained from Thermo Fisher Scientific (Waltham, MA, USA). The model 722 visible spectrophotometer was purchased from Guanze Technology Co., Ltd. (Tianjin, China). The drug concentration analyzer was obtained from Siemens (Shanghai, China).

### 2.3. Synthesis of Fe_3_O_4_ NPs

FeSO_4_·7H_2_O (0.35 g) and FeCl_3_·6H_2_O (0.6 g) were dissolved in 10 mL of deionized water and filtered using a 0.22 μm water system filter membrane. The filtrate was added to a three-necked flask containing 120 mL of deionized water under the protection of N_2_. The mixture was mechanically stirred at 80 °C for 30 min at a stirring speed of 260 rpm. Then, 5 mL of ammonia water was added and stirred at 80 °C for 30 min. After the reaction was completed, the materials were cooled to room temperature and then oscillated with ethanol and deionized water three times. The materials were magnetically separated during washing. The Fe_3_O_4_ obtained was relatively stable and weighed about 200 ± 10 mg after repeated drying and weighing. After subsequent experiments, the optimal dosage of Fe_3_O_4_ was 400 mg. Therefore, drying the synthesized material after each cleaning was not necessary. Deionized water (10 mL) was added into 400 mg Fe_3_O_4_ to make a 40 mg mL^−1^ colloidal solution for later use.

### 2.4. Synthesis of Fe_3_O_4_@SiO_2_

TEOS (2.5 mmol) was dissolved in 1 mL of anhydrous ethanol and mixed as a silicon solution. Then, 10 mL of 40 mg mL^−1^ Fe_3_O_4_ NP colloidal solution, 80 mL of methanol, 20 mL of deionized water, and 1 mL of ammonia solution were mixed in a three-necked flask and were dispersed by ultrasound. The flask was attached to a mechanical mixer and stirred at 450 rpm in a water bath of 30 °C. Afterward, 259 μL of the pre-prepared silicon solution was added per hour six times, and the reaction continued for 6 h. Fe_3_O_4_@SiO_2_ was separated by an external magnetic field, and the obtained solid particles were washed with ethanol and deionized water twice each. The obtained Fe_3_O_4_@SiO_2_ was approximately 490 ± 10 mg after drying, and 40 mL of deionized water was added to make a 12.25 mg mL^−1^ colloidal solution for reserve.

### 2.5. Preparation of MMIPs and MNIPs

Magnetic molecularly imprinted polymers (MMIPs) were synthesized by the sol–gel method. MTX (1 mmol) was dissolved in 80 mL of methanol solution, added to 300 μL of ammonia solution, and the ultrasonically mixed. The solution was placed at room temperature and was mechanically stirred at 260 rpm. After 10 min, 20 mL of 12.25 mg mL^−1^ Fe_3_O_4_@SiO_2_ colloidal solution was added and was further stirred. After 6 min, 1 mL of ammonia solution was added, and 2 mmol APTES was added 15 min later. After 30 min, 6 mmol TEOS was added, and the reaction was finished by stirring at room temperature for 20 h.

The polymers obtained in the three-necked flask were the MMIPs. The MMIPs were washed with methanol solution three times, and then the template molecules in the molecularly imprinted holes were washed with 0.05 M potassium hydroxide solution and methanol (1:1, *v*/*v*). Finally, the polymers were washed with a methanol solution three times again, and the obtained materials were dried at 45 °C for 12 h. Magnetic non-imprinted polymers (MNIPs) were prepared under the same conditions, but without MTX in the synthesis.

### 2.6. Related Adsorption Experiments and Characterization of MMIPs and MNIPs

The MTX–methanol solution was prepared, and appropriate amounts of MMIPs and MNIPs were reserved. Static, dynamic, and selective adsorption experiments were performed using these solutions.

The four polymers (Fe_3_O_4_ NPs, Fe_3_O_4_@SiO_2_, MMIPs, and MNIPs) were characterized by transmission electron microscopy (TEM), Fourier transform infrared (FT-IR) spectroscopy, X-ray diffractometry (XRD), and X-ray photoelectron spectroscopy (XPS).

### 2.7. MSPE Process

Methanol (10 mL) was added with 50 μL of normal serum without MTX, and was ultrasonically mixed and centrifuged at 8000 rpm for 5 min. The supernatant was retained. MMIPs (100 mg) were placed in a 15 mL centrifuge tube, added to 10 mL of methanol, and shaken for 5 min. After complete activation, the liquid was separated and discarded by magnetic separation. The MTX standard solution (400 μL, 0.25 mg mL^−1^) and the supernatant (4.6 mL) were added to the centrifuge tube containing 100 mg MMIPs to a final concentration of 0.02 mgmL^−1^ MTX. The samples were extracted and loaded by shaking at room temperature for 240 min. For the magnetic separation and recovery of MMIPs after sample loading, 5 mL of water–methanol (4:1, *v*/*v*) eluent was added and shaken for 1 min, residual impurities were washed away, and the material was recovered again. Finally, 5 mL of methanol–acetic acid (4:1, *v*/*v*) eluent was added to the treated MMIPs, and the eluent was oscillated for 60 min. After magnetic separation, the liquid was poured into the test tube, dried with N_2_, and redissolved with 500 μL of methanol solution. The chromatograms were detected by HPLC–UV after filtration with a 0.22 μm organic filter membrane.

### 2.8. HPLC–UV Analysis

HPLC–UV (Waters) was used for the MTX detection. The filtered 500 μL complex solution was placed into the liquid injection bottle and into the sample bin. The chromatographic column was C18 (150 mm × 4.6 mm, 5 μm). The parameters were set as follows: the detection wavelength was 306 nm, the flow rate was 1 mL min^−1^, the injection volume was 20 μL, and the temperature was 30 °C. Gradient elution was selected during detection, and the elution conditions are shown in Table 1.

### 2.9. Sample Preparation and Ethics Statement

The clinical samples used in this study were all from the laboratory of the Affiliated Hospital of Qingdao University. Five patients (sample numbers 1–5) were randomly selected for MTX administration. The blood concentration of MTX was detected once on the day of administration using the drug concentration analyzer, and then the residual concentration in the patients was detected according to the time sequence after administration until the result was close to 0. The plasma samples collected from the same patient at different times were numbered in ascending order according to the number of interval days. For example, the samples from patient 1 collected on days 1–4 were numbered as 1-1, 1-2, 1-3, and 1-4, respectively. The numbering rules for other samples were the same as above. The remaining plasma was collected and stored in the refrigerator at −80 °C for later use. This study was approved by the Medical Ethics Committee of the Affiliated Hospital of Qingdao University, and patients were informed and provided consent before blood collection.

The actual samples were tested, and the plasma samples of the patients treated with MTX were divided into two parts. One part was processed by MSPE, and 5 mL of the supernatant was added in the extraction and loading process without adding the standard solution. The leaching and elution operations were the same as the above procedures. The other part was tested using the enzyme multiplied immunoassay technique (EMIT). Finally, Wilcoxon analysis was performed on the test results of the two methods to determine any differences.

## 3. Results and Discussion

### 3.1. Preparation of MMIPs

Fe_3_O_4_ NPs were synthesized by the chemical coprecipitation method, which is easy to operate and costs less, and the synthesized particles have strong magnetic properties [31,32]. In addition, the subsequent polymers were synthesized by the sol–gel method, which consists of two sol–gel processes. One was using TEOS as the precursor and coating silicon shell on the surface of Fe_3_O_4_ NPs to generate Fe_3_O_4_@SiO_2_ with a core–shell structure, and the other was using APTES as the monomer and TEOS as the cross-linking agent. MMIPs and MNIPs were obtained by coating the surface of Fe_3_O_4_@SiO_2_ with a molecular imprinting layer [14]. The preparation process of this experiment is shown in Figure 1. The proportions of the template, monomer, and cross linker can affect the recognition characteristics of MMIPs and MNIPs toward MTX. Therefore, the adsorption capacities of MMIPs and MNIPs for MTX (*C*_MTX_ = 1 mgmL^−1^) prepared using different templates, monomers, and crosslinkers are discussed in this experiment, and the results are shown in Table 2. The results showed that a larger *Q*_MTX_ corresponds to a greater adsorption capacity for MTX, and a larger *Q*_MMIPs_/*Q*_MNIPs_ corresponds to a stronger recognition specificity to MTX [33]. When the ratio of the template, monomer, and crosslinker was 1:2:6, MMIPs had the maximum adsorption capacity for MTX, MNIPs had the minimum adsorption capacity, and *Q*_MMIPs_/*Q*_MNIPs_ had the maximum adsorption capacity. Therefore, the proportion of the template, monomer, and crosslinker selected for the preparation of molecularly imprinted materials was 1:2:6.

The calculation formula of Q_MMIPs_ (Q_MMIPs_) was as follows: Q_MMIPs_ (Q_MMIPs_) = (total methotrexate content in solution — methotrexate adsorption capacity)/extractant quality × 1000.

### 3.2. Related Adsorption Experiments

The static and dynamic adsorption properties of the polymer were investigated under optimal experimental conditions to understand the MTX adsorption capacity of MMIPs. First, 20 mg MMIPs powder and 20 mg MNIPs powder were weighed and placed in a 1.5 mL EP tube, and 0.25 mgmL^−1^ MTX solution was prepared with methanol and diluted gradually into concentration gradients of 0.01, 0.02, 0.05, and 0.1 mg mL^−1^. In this experiment, the static adsorption curves of MMIPs and MNIPs for MTX were measured within the concentration range of 0.01–0.25 mg mL^−1^, and the results are shown in Figure 2A. The amount of MTX bound to the MMIPs increased rapidly with the increase in MTX concentration, and the amount of MTX bound to the MMIPs was considerably higher than that bound to the MNIPs at the same concentration. The result was attributed to the presence of imprinted holes on the MMIP surface, which could specifically absorb MTX; thus, MMIPs were able to absorb more MTX than MNIPs. The dynamic adsorption curve reflects the kinetic process of adsorption, and the adsorption equilibrium time can be obtained by reading the curve. Then, 40 mg MMIPs powder and 40 mg MNIPs powder were weighed and placed in a 15 mL centrifuge tube, and 10 mL 0.25 mg mL^−1^ MTX methanol solution was added to the tube. In the dynamic adsorption experiment, the adsorption capacity of MMIPs and MNIPs for MTX changed with time, as shown in Figure 2B. The binding capacities of the two polymers increased with time, and their adsorption capacities almost reached equilibrium at 60 min. The adsorption curve of MMIPs rose faster, and their adsorption capacity was larger at the adsorption equilibrium, indicating that their adsorption rate was faster and their adsorption capacity was stronger. The results indicate that MTX molecules quickly reached the imprinted holes on the surface of MMIPs; thus, the MMIPs prepared in this experiment had a strong application potential. In addition, the kinetic data of MMIPs and MNIPs were fitted, and finally, the pseudo-second-order model was used to analyze the kinetic data. The model equation and results are shown below:(1)$$\frac{t}{q_{t}}=\frac{1}{k_{2}\cdot q_{e}^{2}}+\frac{t}{q_{e}}=\frac{1}{v_{0}}+\frac{t}{q_{e}}$$
where *q_t_* (mg g−1) and *q_e_* (mg g−1) are the amounts of the MTX adsorbed onto MMIPs and MNIPs at equilibrium time and time *t*, respectively; *k*_2_ (g mg^−1^ min^−1^) is the rate constant of the pseudo-second-order model; and ν_0_ is the initial adsorption rate. The plot of *t*/*q_t_* versus *t* is shown in Figure 2C. It can be clearly seen that a good linear relationship exists between *t*/*q_t_* and *t*, indicating that the pseudo-second-order model fits the experimental data well. Additionally, the correlation coefficients (r^2^) for the pseudo-second-order model at different temperatures are all greater than 0.99, indicating that the pseudo-second-order model provides a perfect fit for the adsorption kinetics of MTX onto the polymers.

Four parts of 20 mg MMIPs powder and four parts of 20 mg MNIPs powder were weighed into 1.5 mL EP tubes. A solution of 0.25 mg mL^−1^ MTX and its structural analogues, metsulfuronmethyl, trimethoprim, and sulfamethoxazole, was prepared with methanol and was diluted to concentration gradients of 0.01, 0.02, 0.05, and 0.1 mg mL^−1^. Figure 3 shows the adsorption of MTX and its structural analogs, metsulfuronmethyl, trimethoprim, and sulfamethoxazole, on MMIPs and MNIPs. The adsorption capacity of MMIPs for the template molecule (MTX) was the largest, whereas its adsorption capacity for non-template molecules was lower. The adsorption effect of MMIPs on MTX and its structural analogs was better than that of MNIPs, because the holes on MMIPs could only specifically bind to MTX. This finding further indicates that MMIPs had a good ability to selectively recognize MTX.

### 3.3. Characterization of MMIPs and MNIPs

The TEM images of the four polymers (Fe_3_O_4_ NPs, Fe_3_O_4_@SiO_2_, MMIPs, and MNIPs) are shown in Figure 4. As shown in Figure 4A, Fe_3_O_4_ NPs was spherical with clearly visible particles with an average diameter of about 15 nm. Figure 4B shows that the surface of the Fe_3_O_4_ NPs particles was coated with SiO_2_. Figure 4C,D shows the electron microscopy images of the MMIPs and MNIPs, respectively. It can be seen that the surface of Fe_3_O_4_@SiO_2_ was successfully coated with molecularly imprinted polymers and had a certain thickness. The polymers were irregular, which was in accordance with the morphological characteristics of the MMIPs and MNIPs synthesized by the sol–gel method.

The XRD spectra of the polymers, which can reflect the crystal structure of the polymer, are shown in Figure 5. The six crystal plane constants of Fe_3_O_4_ corresponded to the peaks at the 2θ angles of 30.10°, 35.51°, 43.15°, 53.59°, 56.95°, and 62.70°, indicating that the elemental composition of Fe_3_O_4_ particles did not change during the preparation process.

Figure 6 shows the XPS spectra of the polymers. The peak at 710.98 eV represents Fe2p, which indicates the presence of Fe_3_O_4_ in all four polymers. The O1s peak at 530.2 eV existed in Fe–O, C=O, C–O, and Si–O–Si. The binding energy peak of Si 2p at 103.17 eV reflected the existence of Si–O–Si, indicating that Fe_3_O_4_ NPs were successfully coated with SiO_2_. In addition, the peak binding energy of 401.84 eV represented N1s, was present in –NH_2_ and indicates the successful synthesis of MMIPs and MNIPs.

### 3.4. Optimization of MSPE Conditions for MTX Enrichment

#### 3.4.1. Selection of Extraction Dosage and Time

This experiment explored the influence of the amount of the extraction agent (MMIPs) on the extraction sample loading effect, and the results are shown in Figure 7A. The peak area of the chromatogram gradually increased when the dosage of MMIPs increased from 20 mg to 100 mg, but did not further increase when the dosage increased to 150 mg. Moreover, maximum adsorption was reached when the MNIP dosage was 50 mg. Therefore, the MMIP dosage of 100 mg was selected to obtain the best MTX loading effect.

In addition, the effect of extraction time on sample loading was further studied. As shown in Figure 7B, the extraction agent achieved the best adsorption effect when the extraction time was 120 min. The peak area of MTX remained stable when the extraction time was more than 120 min. Therefore, 120 min was selected as the optimal extraction time for sample loading in this study.

#### 3.4.2. Selection of Leaching Solution

The selection of a leaching solution should follow two principles. On the one hand, it should minimize the removal of interferent substances retained on the MMIPs. On the other hand, it should ensure the minimum loss of MTX during leaching. In this experiment, water, water–methanol, and water–acetonitrile were used as the leaching solutions, and the MTX concentrations in the supernatant after leaching were measured to explore the effect of different volume ratios of the leaching solution on MSPE. As shown in Figure 8, when the volume ratio of water to methanol was 8:2, the difference between the chromatographic peak areas of MMIPs and MNIPs was the smallest, indicating that the leaching effect of this solution was the best. Therefore, water–methanol (8:2, *v*/*v*) was selected as the leaching solution in this experiment.

#### 3.4.3. Selection of Elution-Related Parameters

In this study, different percentages of methanol–acetic acid solution were used as the eluent to extract the MTX adsorbed on the MMIPs. The recovery rate was the percentage of content of MTX adsorbed and eluted in the extraction process in the total amount. The influence of acetic acid content on MTX extraction recovery is shown in Figure 9A. The extraction recovery rate of methanol–acetic acid (8:2, *v*/*v*) was the largest, this might be the effect of similarity and impermissibility; therefore, methanol–acetic acid (8:2, *v*/*v*) was selected as the eluent for MSPE.

The influence of eluent volume on MTX extraction recovery was investigated, and the results are shown in Figure 9B. The eluent volume increased from 2 mL to 5 mL, and the recovery rate increased from 64.65% to 93.51%. However, the extraction recovery rate did not increase considerably with the further increase in eluent volume. Therefore, 5 mL eluent was selected for elution in this experiment.

Figure 9C reflects the influence of elution time on MTX extraction recovery. Extraction recovery tended to be stable at 60 min and did not further increase. Therefore, 60 min was chosen as the optimal elution time for the MSPE process.

#### 3.4.4. The Regeneration Performance

The regeneration performance of MMIPs was investigated. The extraction process optimized by the above conditions was used to extract methotrexate, and the sample recovery rate of the first experiment was 93.51%. The extractant used was taken out and dried. The second extraction experiment was carried out under the same conditions, and the recovery rate was 88.58%. It can be seen that the recovery rate of the extraction agent prepared in this study only decreased by 4.93% after repeated use for two times, showing a good regeneration performance.

### 3.5. Analytical Characterization

Firstly, the standard curve of MTX spiked solution detected by the MSPE–HPLC–UV method was drawn in this experiment, as shown in Figure 10. The results showed that the MTX measured by the spiked method was linear in the range of 0.00005–0.25 mg/mL, and the linear equation was Y = 43242.85C (mg/mL)−0.90178 (R^2^ = 0.9903). According to the signal-to-noise ratio (S/N = 3), the detection limit of the sensor was 12.51 ng/mL. The detection limit of this method was lower than the spiked concentration of MTX, which could meet the needs of the MTX blood concentration detection below and enrich the detection method of MTX to a certain extent.

To evaluate the superiority of the prepared MSPE–HPLC–UV method, a comparison of this work with other reported methods for MTX determination is shown in Table 3. From Table 3, it is apparent that the superiority of this work was quite satisfactory.

### 3.6. Human Plasma Analysis

The established MSPE–HPLC–UV method was used to detect the MTX content of the actual plasma samples, and the detection results were compared with those of EMIT, as shown in Table 4. Wilcoxon analysis was performed on the test results of the two methods, and the results are shown in Table 5. The results showed that the detection results of the two methods had no significant difference (*p* = 0.360). Therefore, the MSPE–HPLC–UV method established in this experiment had a good accuracy for MTX detection.

The chromatograms of the MTX standard solutions were compared with those of MMIPs and MNIPs at the same concentration. As shown in Figure 11, the characteristic peak of MTX was at 13 min. The chromatogram baseline of the sample extracted by MMIPs was smooth, and the chromatogram peak type and peak area were similar to those of the MTX standard solution with the same concentration. The peak area of the MNIP extraction was much smaller than that of the standard solution. The results indicate that MSPE–HPLC–UV method can resist interferent substances in the plasma and can enrich MTX. Moreover, the MTX detection method established in this experiment had a high accuracy and good practicability and could be used for MTX determination in actual plasma samples.

## 4. Conclusions

In summary, MMIPs as a new MTX adsorption material were prepared. A HLPC detection method based on magnetic molecularly imprinted solid-phase extraction was established to achieve the efficient adsorption and specific recognition of MTX in the patients’ plasma. It simplified the sample pretreatment process, reduced the interference caused by protein and other components in the plasma to MTX detection, and laid a foundation for the determination of the MTX blood concentration. This study proved that the proposed method can directly detect the serum MTX concentration, which provides a new method of support for the detection and monitoring of the MTX concentration, and enriches the application of magnetic materials and molecularly imprinted materials in clinical medicine.

## Figures and Tables

**Figure 1 molecules-27-06084-f001:**
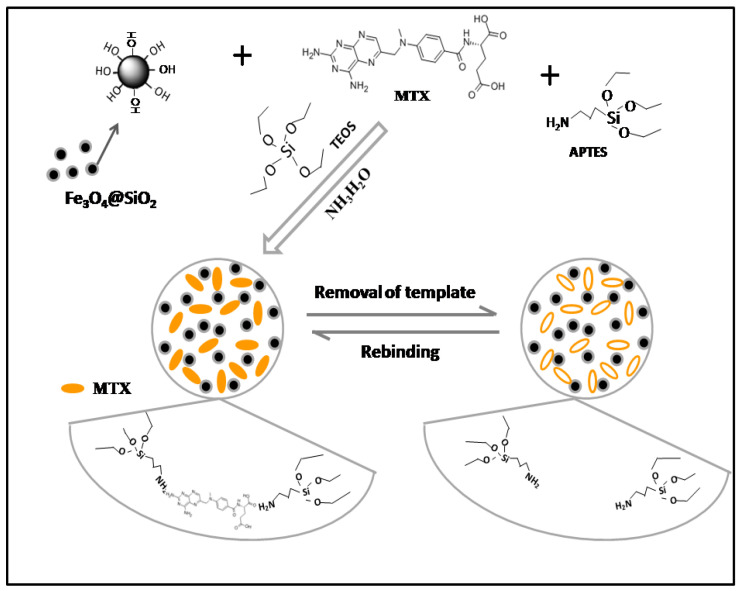
Preparation of MMIPs via the sol–gel process.

**Figure 2 molecules-27-06084-f002:**
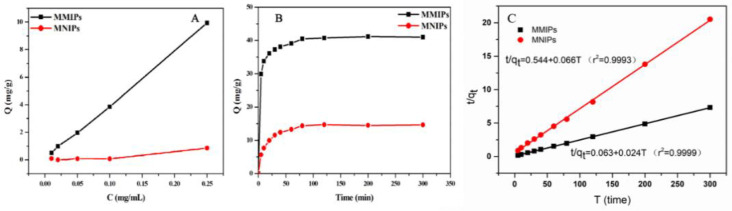
(**A**) Static adsorption curves of MMIPs and MNIPs; (**B**) dynamic adsorption curves of MMIPs and MNIPs; (**C**) dynamic adsorption curves fitted by the pseudo-second-order model.

**Figure 3 molecules-27-06084-f003:**
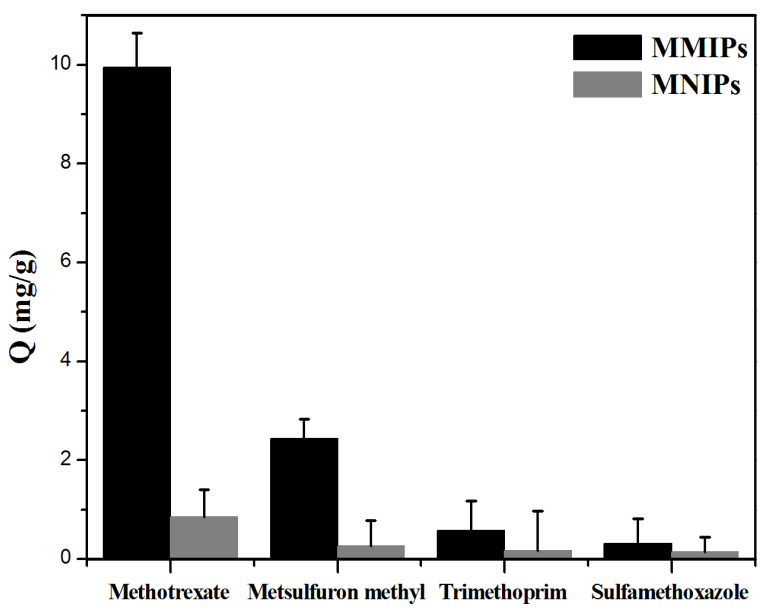
Adsorption performance of MMIPs and MNIPs on MTX and its structural analogues (0.25 mg mL^−1^).

**Figure 4 molecules-27-06084-f004:**
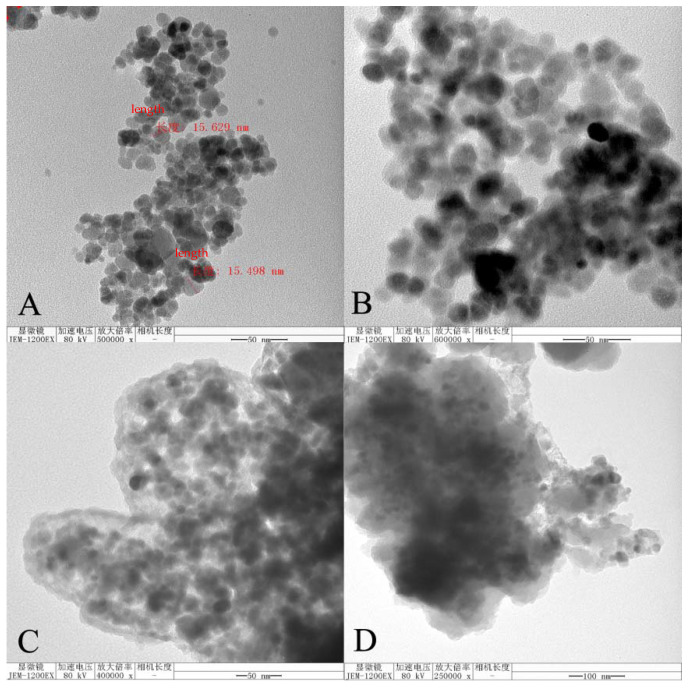
Transmission electron microscopes of (**A**) Fe_3_O_4_ NPs, (**B**) Fe_3_O_4_@SiO_2_, (**C**) MMIPs, and (**D**) MNIPs.

**Figure 5 molecules-27-06084-f005:**
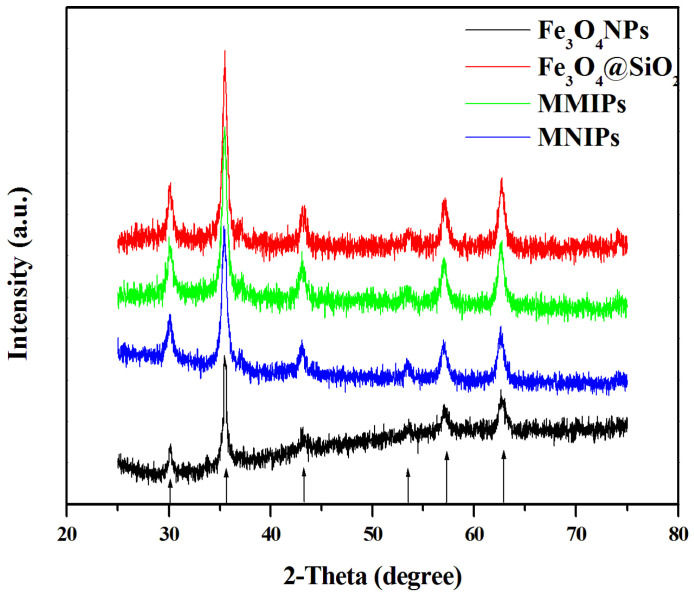
XRD spectra of the four polymers (Fe_3_O_4_ NPs, Fe_3_O_4_@SiO_2_, MMIPs, and MNIPs).

**Figure 6 molecules-27-06084-f006:**
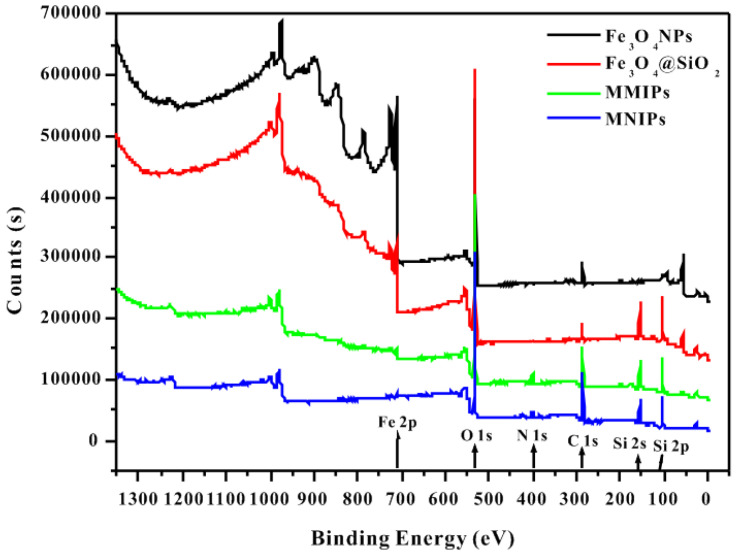
XPS spectra of the four polymers (Fe_3_O_4_ NPs, Fe_3_O_4_@SiO_2_, MMIPs, and MNIPs).

**Figure 7 molecules-27-06084-f007:**
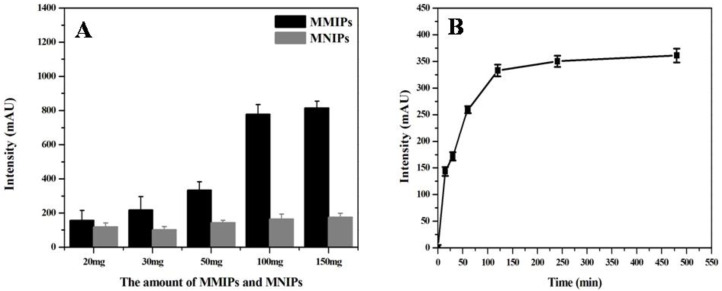
Optimization of conditions affecting the extraction effect of MTX on the MMIPs, including (**A**) the amount of MMIPs and MNIPs, and the (**B**) extraction time.

**Figure 8 molecules-27-06084-f008:**
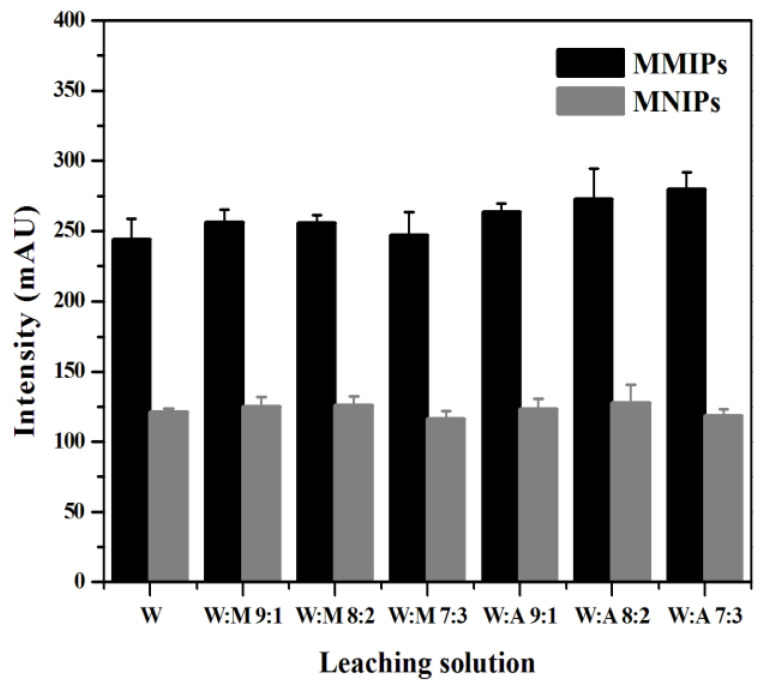
Optimization of conditions affecting the extraction effect of MTX on the MMIPs in regard to the leaching solution.

**Figure 9 molecules-27-06084-f009:**
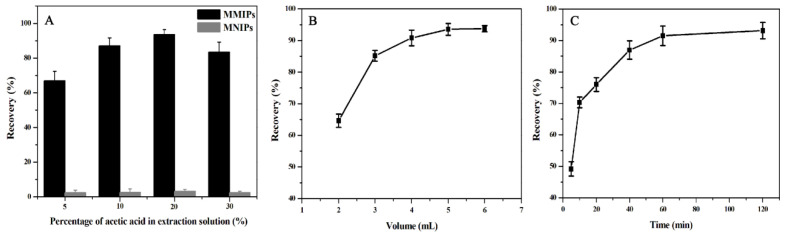
Optimization of conditions affecting the extraction effect of MTX on the MMIPs, including (**A**) percentage of acetic acid, (**B**) eluent volume, and (**C**) elution time.

**Figure 10 molecules-27-06084-f010:**
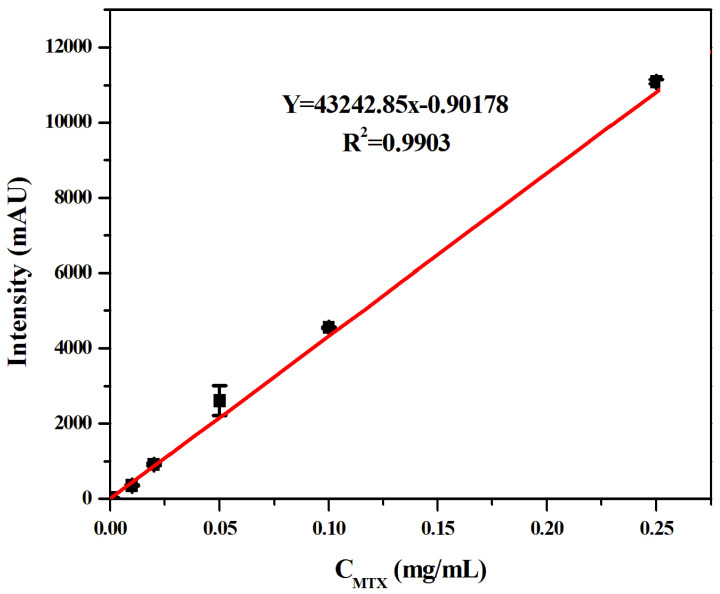
Linear curves between MTX liquid chromatography peak area and its concentration.

**Figure 11 molecules-27-06084-f011:**
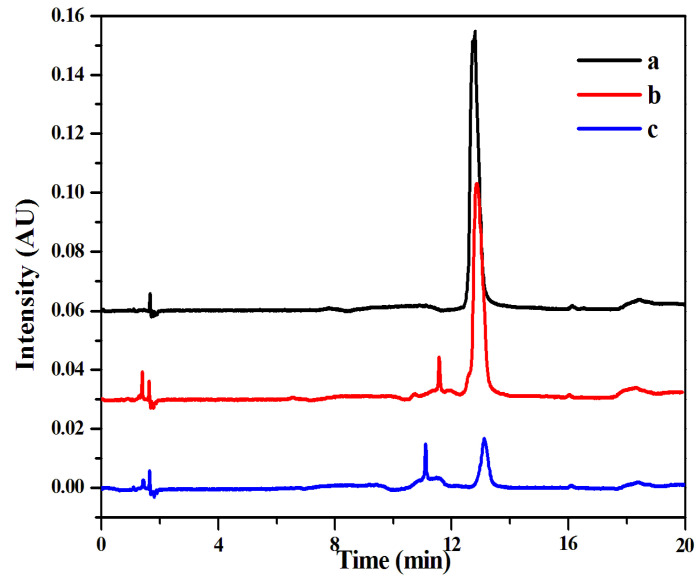
High performance liquid chromatography of the three solutions: MTX standard solution ((a) 0.05 mg mL^−1^), the spiked solution extracted by MMIPs ((b) 0.05 mg mL^−1^) and the spiked solution extracted by MNIPs ((c) 0.05 mg mL^−1^).

**Table 1 molecules-27-06084-t001:** Gradient elution conditions for HPLC–UV detection.

Time (min)	Water (%)	Methanol (%)
0	90.0	10.0
5	90.0	10.0
10	0.0	100.0
15	0.0	100.0
20	90.0	10.0

**Table 2 molecules-27-06084-t002:** The influence of the ratio of the template, monomer, and crosslinker on the adsorption capacity of the polymers.

Template: Monomer: Crosslinker (mM)	Q_MTX_ (mg g^−1^) (MMIPs)	Q_MTX_ (mg g^−1^) (MNIPs)	Q_MMIPs_-Q_MNIPs_ (mg g^−1^)	Q_MMIPs_/Q_MNIPs_
1:2:4	36.91	21.32	15.59	1.73
1:2:6	39.56	4.21	35.35	9.40
1:2:8	16.49	5.66	10.83	2.91
1:2:10	12.28	6.23	6.05	1.97
1:3:6	33.19	19.97	13.22	1.66
1:4:6	34.64	46.24	−11.60	0.75
1:5:6	22.33	45.23	−22.90	0.49
1:6:6	9.28	26.04	−16.76	0.36

**Table 3 molecules-27-06084-t003:** A comparison between this work and other reported works.

Recognition Element	Linearity Range	Method LODs	References
N, S co-doped CQDs	0.4–41.3 μg/mL	12 ng/mL	[34]
APTES-CPDs	2.0–50 µg/mL	2.0 µg/mL	[35]
Surface-Enhanced Raman Scattering	5–150 μmol/L	2.1μmol/L	[36]
MSPE-HPLC-UV	0.00005–0.25 mg/mL	12.51 ng/mL	This work

CQDs: carbon quantum dots. APTES-CPDs: amine-functionalized silica carbon polymer dots.

**Table 4 molecules-27-06084-t004:** Comparison of the MTX concentration in the plasma between EMIT and MSPE–HPLC–UV.

	EMIT	MSPE-HPLC-UV
Sample	Found (Μm)	Found (μM)
1-1	0.70	0.536
1-2	0.65	0.519
1-3	0.26	0.307
1-4	0.21	0.196
2-1	0.36	0.267
2-2	0.16	0.216
2-3	0.11	0.111
3-1	5.00	4.641
3-2	0.73	0.756
3-3	0.36	0.306
3-4	0.18	0.169
4-1	23.00	20.585
4-2	6.60	6.856
4-3	3.40	3.493
5-1	38.40	42.135
5-2	34.80	36.278
5-3	33.40	29.150
5-4	0.10	0.119

**Table 5 molecules-27-06084-t005:** Wilcoxon analysis of the MTX plasma concentration by EMIT and MSPE–HPLC–UV.

Method	Median Difference	Z Value	*p*
EMIT vs. MSPE–HPLC–UV	0.147	0.915	0.360

## Data Availability

The date was generated during the study.

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
