# Peer review of "Magnetic Molecularly Imprinted Polymers for the Rapid and Selective Extraction and Detection of Methotrexatein Serum by HPLC-UV Analysis"

_molecules, 2022, doi:10.3390/molecules27186084_

Round 1
Reviewer 1 Report
The submitted manuscript reported the magnetic molecularly imprinted polymers for the rapid and selective extraction and detection of methotrexate. The topic is interesting, and the proposed method could do a favor to the detection technology with specific selectivity. I, therefore, think the manuscript can be considered for publication after addressing the issues mentioned below:
1. Abstract. Authors should mention the innovation of the work in the abstract.
2. Introduction. Since molecular imprinting technology (MIT) is the core technology of this work, its development in separation, purification, and detection should be discussed, such as 10.1002/adma.202100543, 10.1016/j.memsci.2022.120750, 10.1016/j.cej.2022.137825, 10.1016/j.cej.2019.122309, and so on.
3. Section 2.1. The purity of the chemicals should be mentioned.
4. Section 3.3. It is not sufficient to infer the formation of Fe-O, C=O, C-O, Si-O-Si and -NH2 just based on the presence of O, Si and N elements in the wide-scale XPS spectra. The authors should provide and analyze O 1s, Si 2p and N 1s spectra on a narrow scale with split peaks.
5. Section 3.4.3. The authors mentioned that the extraction recovery rate of methanol-acetic acid (8:2, v/v) was the largest. Possible reasons for this phenomenon should be speculated. Which effect leads to the phenomenon? Protonation or the "similarity and intermiscibility" effect?
6. The regeneration performance should be investigated.
7. A comparison between prepared product and reported work, for example, a table, should be carried out to exhibit the superiority of this work.
Reviewer 2 Report
To the authors:
Manuscript ID molecules-1889257 entitled “Magnetic molecularly imprinted polymers for the rapid and selective extraction and detection of methotrexate in serum by HPLC-UV analysis” is carefully organized. The authors used methotrexate as template to carry out the study of small molecular imprinting. Magnetic solid-phase extraction and liquid chromatography were used for separation, enrichment and quantification, which further improved the detection limit of the analytical method. However, there are still many issues that need to be further improved.
Some comments are as follows:
1. In section 2.5:
1) Line141-144: 0.05 M potassium hydroxide solution and methanol (1:1, v/v) were used to remove your template molecule MTX. However, in MSPE process, methanol–acetic acid (4:1, v/v) eluent was selected (line 165). Why choose two different eluents to elute the same substance?
2) Line 138-139: As you mentioned, “after 30 min, 6 mmol TEOS was added, and the reaction was finished by stirring at room temperature for 20 h”, why the imprinting time is so long?
2. In section 2.6:
The experimental conditions in static, dynamic and selective adsorption experiments should be pointed out more clearly. Or it should be supplemented in section 3.2, such as addition concentration and volume of MTX and its structural analogs.
3. In section 2.8:
According to your description, the liquid phase detector you are using is UV detector. The ordinate of the obtained chromatogram shall be “Intensity”. However, in Figure 7, 8 and 10, the Y-axis of the images were absorbance. The absorbance cannot be read from HPLC, please check.
4. In section 3.1:
1) Line 214: The calculation formula of adsorption amount (QMMIPS and QMNIPS) should be pointed out. The amount of MTX added should be indicated.
2) Imprinting time is also an important factor affecting imprinting effect. Data for optimizing the imprinting time should be given.
5. In section 3.2:
1) Line 228: The concentration of template molecules is 1 mmol (line 134), but in the static adsorption experiment, only the concentration range of 0.01-0.25 mg/mL (line 225-233) was measured. How do you determine the concentration range? Is this the maximum adsorption capacity of MMIPs? I think the concentration range should be wider.
2) Line 234-243: In dynamic adsorption experiment, the reason why the binding ability/ adsorption rate of MMIPs changes at different times should be discussed more deeply. Perhaps you can use different adsorption model equations for fitting analysis.
6. In section 3.4.3 and section 3.5:
1) Line 323-338: The calculation formula of the recovery rate should be pointed out. Moreover, the analysis method needs to be improved, such as giving linearity and range.
7. References:
Some references are not aligned.
8. The English language needs to be improved, and the details of writing should be noted. The analysis of experimental results should be deeper.
Reviewer 3 Report
Minor revision is required before acceptance.
1. The full name of MTX needs to appear once before the abbreviation can be used.
2. To be honest, it's hard for me to see valuable information from TEM.
3. The results look pretty good. I'm a little concerned about the selectivity of MMIP.
Round 2
Reviewer 1 Report
The authors improved their manuscript after revision. Therefore, it can be accepted in the current form.